# Health-Related Behaviors in Adolescents Mediate the Association between Subjective Social Status and Body Mass Index

**DOI:** 10.3390/ijerph17197307

**Published:** 2020-10-07

**Authors:** Asborg A. Bjertnaes, Catherine Schwinger, Petur B. Juliusson, Tor A. Strand, Mads N. Holten-Andersen, Kjersti S. Bakken

**Affiliations:** 1Department of Pediatrics, Lillehammer Hospital, Innlandet Hospital Trust, Anders Sandvigs Gate 17, 2609 Lillehammer, Norway; Madsn.holten-andersen@sykehuset-innlandet.no; 2Department of Clinical Medicine, Faculty of Medicine, University of Oslo, P.O. Box 1171 Blindern, 0318 Oslo, Norway; 3Department of Global Public Health and Primary Care, Centre for Intervention Science in Maternal and Child Health, University of Bergen, 5020 Bergen, Norway; C.Schwinger@uib.no (C.S.); tors@me.com (T.A.S.); 4Department of Health Registries, Norwegian Institute of Public Health, P.O. Box 973 Sentrum, 5808 Bergen, Norway; Petur.Juliusson@uib.no; 5Department of Clinical Science, University of Bergen, Jonas Lies vei 87, 5021 Bergen, Norway; 6Department of Pediatrics, Haukeland University Hospital, P.O. Box 1400, 5021 Bergen, Norway; 7Department of Research, Innlandet Hospital Trust, Furnesvegen 25, 2380 Brumunddal, Norway; 8Department of Gynecology and Obstetrics, Lillehammer Hospital, Innlandet Hospital Trust, Anders Sandvigs Gate 17, 2609 Lillehammer, Norway; kslett@sykehuset-innlandet.no

**Keywords:** adolescents, body mass index, health behavior, obesity, subjective social status, structural equation modeling

## Abstract

The aim of this study was to explore the association between adolescent subjective social status (SSS) and body mass index (BMI) at two different time points and to determine whether this association was mediated by health-related behaviors. In 2002 (n = 1596) and 2017 (n = 1534), tenth-grade students (15–16 years old) in schools in the District of Oppland, Norway, completed a survey. Four categories of perceived family economy were measured as SSS, and structural equation modeling was performed, including a latent variable for unhealthy behavior derived from cigarette smoking, snuff-use, and alcohol-drinking as well as dietary and exercise as mediators. No linear association was found between SSS and BMI in 2002 (standardized ß −0.02, (95% confidence interval (CI) −0.07, 0.03)). However, an association was present in 2017 (standardized ß −0.05 (95% CI −0.10, −0.001)), indicating that BMI decreased by 0.05 standard deviations (0.05 × 3.1 = 0.16 BMI unit) for every one-category increase in SSS. This association was mediated by exercise (standardized ß −0.013 (95% CI −0.02, −0.004) and unhealthy behavior (standardized ß −0.009 (95% CI −0.002, −0.04)). In conclusion, a direct association between SSS and BMI was found in 2017 in this repeated cross-sectional survey of 15–16-year-old Norwegian adolescents. This association was mediated through health-related behavior.

## 1. Introduction

Overweight and obesity (OWOB) in adolescence represent a threat to present and future health [1,2]. Current prevalence of overweight and obesity in adolescents in Norway has been registered as 25% in girls and 24% in boys, and treatment of OWOB has resulted in few long-term success stories [3,4]. Therefore, it seems necessary to focus on risk factors for OWOB to develop prevention strategies.

Lower sociodemographic status has repeatedly been found to be associated with a risk for increased Body Mass Index (BMI) in adolescents, and the mechanism of this association is a matter of discussion [5]. A leading hypothesis in high income countries is that lower sociodemographic status is associated with more unhealthy behaviors, such as a higher consumption of unhealthy nutrition and a lower frequency of exercise [6].

The association between sociodemographic status and BMI in adolescence has been found as dynamic, and studies have revealed both non-existing, negative and positive associations [7,8,9]. During recent decades, food and drinks that are high in calories and low in nutrition have become easier accessible [10]. Furthermore, the need for physically demanding work and transportation has declined [11]. These changes might have affected the association between sociodemographic status and BMI over recent decades [8]. However, the association between adolescent sociodemographic status and health outcomes including OWOB has been found to be complex and in need of further elucidation [12].

Adolescence is one of the critical time points for the individual health trajectory, as OWOB has a high risk of continuing into adulthood [2], and many health-related habits are established during adolescence [12]. Furthermore, behavior in this age group is unlike that of children and adults, among others, as a result of a rapid development of the central nervous system, and also a late maturation of the prefrontal cortex of the brain [12,13]. As the pre-frontal cortex is the location for risk assessment, planning, organization, and delay of pleasure, the adolescent brain is hypersensitive to reward such as the anticipation of food, money, drugs, and social interactions [13,14,15,16]. Accordingly, behavior in this group should be studied separately from other age-groups, and a broader pattern of covariations of unhealthy behaviors should also be explored. If these behaviors are revealed as part of the complicated behavioral patterns that connect increasing sociodemographic status to decreasing BMI, a more comprehensive understanding of this association could be gained.

The understanding of the association between sociodemographic status and BMI in adolescents could also be enhanced by using other measures than the traditionally used objective measures income, education, and occupation. These measures are usually not yet accomplished by adolescents, and another measure, namely, subjective social status (SSS), has been found to be related to both physiological and psychological parameters [17]. Moreover, SSS has also been found to reveal information not captured by objective sociodemographic measures [18].

Overweight and obesity in adolescents is a global problem [19]. Thus, analyses should be done to understand more of the mechanisms that influence increased BMI in adolescents. The a priori hypothesis was that the association between SSS and BMI was mediated by health-related behaviors. Therefore, the aim was to investigate the possible relationship between SSS and BMI in adolescents at two different time points and to determine whether the association was mediated by health-related behaviors in cross-sectional samples of 15–16-year-old adolescents. An analysis was performed to explore whether unhealthy nutrition, lower amount of exercise, and other unhealthy habits mediated the association between sociodemographic status and BMI.

## 2. Materials and Methods

### 2.1. Protocol and Subjects

Tenth-grade students (15–16 years old) in lower secondary schools in Oppland County, Norway, answered a cross-sectional survey in the period April–June 2002 and April–May 2017. Oppland County is predominantly rural and is one of 18 counties in Norway, with a total population of 183,000 in 2002 and 189,000 in 2017. It includes several towns, two of which had populations between 25–30,000 during this period. Although Oppland is a predominantly rural county, Norway is considered a high-income, egalitarian welfare country with relatively small differences between the counties. The survey contained questions on the perceived economic status of the family, nutrition, leisure-time sports, cigarette smoking, snuff-use, alcohol-drinking, and current weight and height. The questionnaires used in 2002 and 2017 were piloted among 10th-grade students [20]. In 2002, the Norwegian Institute of Public Health conducted the study; in 2017, our research team conducted the study in collaboration with the County Governor of Oppland. Participation was voluntary, and written consent was obtained from students above the age of 16 years and from the parents of students younger than 16 years. The survey was carried out in all 46 schools in Oppland in 2002 and in 43 schools (excluding three private schools, accounting for 24 students) in 2017. In 2017, the same three pediatric nurses were present in all school classes to assist and answer questions while completing the survey. A total of 1877 students completed the survey in 2002 and 1793 in 2017. The study sample used in the SEM analyses comprised 1596 participants (77%) in 2002 and 1534 participants in 2017 (69%) due to lack of parental consent and missing data on one or more of the variables (Figure 1).

The Regional Committee for Medical Research Ethics Southeast approved this project (2017 project number: 2016/1755).

#### 2.1.1. Outcome Variables

Based on self-reported weight (to the nearest kg) and height (to the nearest centimeter), the participants’ BMI was calculated (kg/m^2^).

#### 2.1.2. Exposure Variable (SSS)

All participants were asked about their perceived economic situation in the family in comparison to other families. This question had four categories: poor, average, good, and very good.

### 2.2. Statistical Analyses

The level of significance was set to 5%, although it was not interpreted as a definite cut-off, in line with the current statistical and epidemiological understanding of the issue [21]. Missing observations were addressed by listwise deletion, meaning that cases with one or more missing observations were excluded. The participants are described by year of survey (2002 and 2017) in Table 1.

Next, the association between SSS and BMI was estimated using linear regression models. Table 2 displays mean BMI by family economy, with average family economy used as the reference group. As no linear association was found between the exposure and the outcome in the 2002 model, (standardized ß −0.02, (95% confidence interval (CI) −0.07, 0.03)), no model was built containing the indirect associations or a latent variable for this timepoint.

Spearman’s correlation coefficients with corresponding p-values were calculated between all variables included in the SEMs (Table 3). Tables displaying all variables explored during the model-building and their Spearman’s correlation coefficient are included in the Appendix A as Table A1 and Table A2.

#### 2.2.1. SEM Analyses

The a priori hypothesis was that the association between SSS and BMI was mediated by health-related behaviors. The models were also run using gender and age adjusted BMI (BMI z-scores) as the outcome variable to determine whether the results were different when adjusting for gender and age [22].

The model was built with standardized estimates and was performed in two steps:

##### Latent Variables

Confirmatory factor analysis was used to build the measurement model, i.e., the part of the model that builds the latent variable [23]. The following latent variables were explored: a diet high in sugar (consumption of cakes and candy, sugar-sweetened carbonated soda, lemonade and energy drinks), level of exercise (weekly hours of leisure-time workout, member of sports team and mean screen time on school-days), and unhealthy behavior (cigarette smoking, snuff-use, alcohol-drinking, and frequency of brushing teeth). The variables were removed if they did not correlate to the latent variable, had factor loadings <0.4, produced impossible cases (the presence of non-possible values), or resulted in a poor model fit when included. The latent variable should also include at least three or more observed variables [24]. One such latent variable including the variables cigarette smoking, snuff-use, and alcohol-drinking was identified. Table 1 displays the variables used in the models. Figure 2 displays the latent variable marked by a circle and the observed variables marked by a square.

##### Structural Model

To explore the hypothesized direct and indirect effects, a structural model estimating the associations among latent and observed variables was built [23]. The direct association was defined as the path between SSS and BMI. The indirect associations were defined as the paths connecting the SSS and BMI through the variables describing health-related behavior (Figure 2). After identifying one latent variable, exercise and consumption of sugar-sweetened soda were defined as mediators, as they represent central health related behaviors that are risk-factors for OWOB. Thus, only one structural model was built. The mediated proportion of the indirect effect was calculated from the indirect effect/total effect (direct + indirect effect).

##### Model Fit

The a priori decision was to test the goodness of fit of both models using Comparative Fit Index (CFI), Root Mean Square Error of Approximation (RMSEA), Chi-square, and standardized Root Mean Square Residual (SRMR) as indicators [24]. Post hoc modification of the model was not performed.

The excluded and included cases are compared to explore whether the missing observations were missing at random (Table A3).

The SEM analyses were performed in R version 3.6.1 (5 July 2019)—R Core team (2020). R: A language and environment for statistical computing. R Foundation for Statistical Computing, Vienna, Austria. URL https://www.R-project.org/. The libraries lavaan [25], semPlot [26], and semTools [27] were used. Due to the use of ordinal data, the model parameters were explored by the preferred estimator diagonally weighted least squares, including variance-adjusted robust mean and standard errors [24]. For all other analyses STATA 15.0 software (STATA, College Station, TX, United States: StataCorp, 2017) was used.

## 3. Results

The mean age (SD) of the participants was 15.9 years (0.3) in 2002 and 15.8 years (0.3) in 2017 (Table 1). The proportion of boys was 51.3% in 2002 and 47.8% in 2017. The mean BMI increased from 21.1 in 2002 (95% confidence interval (CI) 21.0–21.2) to 21.4 (95% CI 21.3–21.6) in 2017. In 2017, more adolescents reported having good and very good family economy, and fewer adolescents reported having average and poor family economy. More adolescent also reported having healthier habits in 2017. The distribution of the main exposure variable (SSS) and all observed variables are displayed in Table 1.

An approximately normal distribution was found for BMI (skewness and kurtosis 1.21 and 2.81 for 2002, 1.14 and 2.44 for 2017).

The results from the crude linear regression models are displayed in Table 2. When using average family economy as the reference, the mean BMI was only significantly higher in the lowest category poor economy, which included 57 adolescents in the data from 2002. In 2017, the mean BMI was decreased for both groups good and very good family economy when using average family economy as the reference group.

Table 3 presents a correlation matrix between all measured variables used in the SEM analysis. The highest correlations were between snuff-use and cigarette smoking (0.68) and between snuff-use and ever tried alcohol (0.32).

All the observed variables used in the latent variable were coded so that an increasing value indicated a higher use or consumption and thus more unhealthy behavior. Consumption of sugar-sweetened carbonated sodas and hours of leisure-time weekly work out were included as mediating variables and coded as follows: higher values of consumption of sugar-sweetened soda revealed a higher consumption, and higher values of leisure-time work-out revealed more hours working out.

Questions were preferably included if containing more than 4 categories of answers, as this is most suitable when using SEM [24]. Still, ever tried alcohol was chosen even if it had only two categories of answers. This choice was made as the question was considered accurate in addressing the health-related behavior of drinking alcohol or not.

Figure 2 displays the hypothesis, i.e., that the association between SSS and BMI was mediated by health-related behaviors. The model revealed a good fit with a CFI of 0.99, RMSEA of 0.046 (95% CI 0.03, 0.06), and SRMR of 0.04. The chi-square test for the model was significant at *p* < 0.001.

The model explained 2% of the variance in adolescent BMI in 2017, and there was a significant association between SSS and BMI (standardized ß −0.05 (95% CI (−0.10, −0.001)), indicating that the BMI decreased by 0.05 standard deviations (0.05 × 3.1 = 0.16 BMI units) for each one-category increase in SSS (i.e., from average to good perceived family economy) (Figure 2).

This association was partially mediated by the latent variable unhealthy behavior (standardized ß −0.009, (95% CI −0.002, −0.04) and hours of weekly leisure-time workout (standardized ß −0.013, (95% CI −0.02, −0.004)). Thus, a higher SSS category was associated with a lower BMI through both the direct and mediated pathways. Unhealthy behavior mediated 15.3% of the total effect, and the hours of weekly workout mediated 20.6% of the total effect.

When comparing the included and excluded observations due to the use of listwise deletion, only minor differences between them were found (Table A3, Appendix A). Additionally, the model was built using BMI z-score as an outcome which displayed only minor differences (data not shown).

## 4. Discussion

### 4.1. Main Results

In repeated cross-sectional surveys of 15–16-year-old Norwegians, a linear association between SSS and BMI was found in 2017 but not in 2002. In 2017, this association was mediated by frequency of exercise and unhealthy behavior, including cigarette smoking, snuff-use, and alcohol-drinking.

### 4.2. The Association between SSS and BMI in 2002 and 2017

The use of SSS seems adequate to depict adolescent sociodemographic status, as adolescents have not yet finished their education towards achievement of occupation and income. Furthermore, SSS and objectively measured socioeconomic status have been found to be moderately connected [17]. Unique aspects of the association between sociodemographic status and health outcomes have been found using SSS [17]. This measure has successfully been used in studies exploring subjective perception of family economic status and BMI in adolescents, subjective perception of rank within a school hierarchy and obesity in adolescents, and perceived rank within society and body fat distribution in female Caucasian adults [17,28,29]. Accordingly, several nuances of the association between OWOB and SSS have been revealed, and the association seems valid.

A significant linear association between SSS and BMI was found in 2017 but not in 2002, which may reflect a time trend for this association in the Norwegian setting. A shift in the association is described in several reviews among child- and adolescent populations in developed countries: A review published in 1989 found that 26% of the studies revealed an association between higher sociodemographic status and obesity [30]. A review published in 2008 found that the positive association between sociodemographic status and OWOB had almost disappeared [31]. Finally, a review from 2015 found that higher weight was associated with lower sociodemographic status [32]. This development is further supported by a longitudinal study from the same time period in the UK [9].

The finding of different social patterns of OWOB can be put in context through the epidemiological transition, describing predominating patterns of morbidity and mortality, including OWOB [8,33]. Population groups with more resources gain access to more food first, and thus, OWOB can be a sign of wealth [8]. Subsequently, the “western” lifestyle and living standards are achievable for the less economically privileged, thus increasing access to unhealthy food and possibilities for overfeeding as well as facilitating a less physically active lifestyle. Hence, a higher prevalence of OWOB will be found in the lower-income groups, possibly connected to cheaper energy-dense foods [8,34].

In general, several mediators can be included when investigating the association between SSS and BMI. Differences in nutrition, physical activity, the built environment, and genes have been discussed [35,36,37]. It could also be speculated whether the psychological resources in adolescents including social capital have a mediating role in the association between sociodemographic status and BMI, as found in adults [38,39].

### 4.3. Unhealthy Behaviors in Adolescents

A correlation was found between cigarette smoking, snuff-use and alcohol-drinking in adolescents. This finding is in line with a study that revealed adolescent health-related behavior as a continuum from preventive health behaviors to unhealthy behaviors [40]. The co-variation of the unhealthy behaviors in adolescents seen in our study has been described as an age-typical pattern of behavior which is possibly influenced by an immature cognitive control system [13]. Thus, it seems reasonable to address behavior in adolescents separately from other age-groups.

The relationship between sociodemographic status and BMI is dynamic and has been found as both positive and negative depending on the country’s developmental status [8]. Therefore, it seems reasonable to explore the societal frames for health-related choices as mediators of this association. Besides the finding of a direct association between SSS and BMI in 2017, the SEM explored three indirect associations between these variables. First, the finding that more exercise mediated the negative association between SSS and BMI is in line with an earlier review in European children and adolescents [36]. Second, the association between SSS and BMI was not mediated by consumption of sugar-sweetened carbonated sodas, which is inconsistent with previous research [41,42]. This may partly be influenced by that the sales figures for sugar-sweetened carbonated soda in the last decade have declined by about 20%. Simultaneously, the numbers for sugar-free carbonated sodas have inclined and reached comparable numbers with sugar-sweetened carbonated sodas [43]. Thus, there is a possibility that sugar-sweetened soda has lost its position as a main driver of the obesity epidemic. The decrease in consumption of sugar-sweetened carbonated soda was also revealed in our data for 2002 and 2017 (Table 1). Another possibility is the use of a non-validated nutrition-record in our study. Third, cigarette smoking, snuff-use, and alcohol-drinking were found as highly correlated, and the latent variable that included those habits mediated the association between SSS and BMI in 2017. The associations between these behaviors and sociodemographic status have earlier been found diverging, as use of nicotine products like cigarette smoking and snuff-use have been found associated to lower sociodemographic status, while alcohol-drinking has shown a more complex pattern related to higher sociodemographic status in some countries in Europe [44,45,46,47]. Another systematic review of cigarette smoking, unhealthy nutrition, alcohol consumption, and less exercise in adolescents and adults found that these behaviors tended to cluster with lower sociodemographic status, and also with younger age [48].

In our study, hours of exercise mediated 20.6% of the association between SSS and BMI, and unhealthy behavior mediated an additional 15.3% of this association in adolescents. The finding can be interpreted as that the SSS is associated with BMI in a manner where SSS influences habits exceeding those directly related to an energy sur-plus. This interpretation is in line with the established perception that sociodemographic status also involves capabilities that surpass obtaining services and goods that promote health [6]. This new information can contribute to the understanding of how SSS influences BMI.

As adolescence is a critical phase for later health, it is important to explore new information regarding the association between SSS and BMI [2,12]. The finding of not only physical activity but also unhealthy behavior as a mediator in this association can possibly suggest why it is so difficult to both prevent and treat OWOB: Adolescents have multiple health-related every-day habits that influence BMI. Thus, it seems important to communicate the importance of limiting an obesity-enhancing environment to policymakers.

The finding of lower sociodemographic status as a risk factor for increased BMI also has implications for prevention efforts. An earlier review found that interventions aiming at obesity prevention had different effects based on the person’s sociodemographic status [49]. Thus, lower sociodemographic status is a barrier to prevention of increased BMI should be communicated to policymakers.

### 4.4. Strengths

The strengths of this study include that the repeated cross-sectional studies reflect a general adolescent population, and the sample sizes are relatively large. The survey was repeated in the same district, at the same time of year, and at the same participant age. Accordingly, the numbers are seemingly comparable.

Using SEM analysis allowed assessment of indicators of health-related behavior, including a latent variable of unhealthy behavior as a mediator in the association between SSS and BMI. This method of analyses provided a more comprehensive measurement compared to using the variables as separate indicators. This method also quantifies the measurement errors and unexplained variances and provides a measure of how well the model fits the covariance in the dataset [50]. Further, a reflective measurement model was used when building the latent variable [51]. Latent variables can be thought of as a hypothetical construct that reflect a not directly observable co-variating pattern, such as e.g., personality [51]. Latent variables might represent a structure otherwise observed as a clustering behavior [52].

The model-fit is considered as good; still, the theory behind the model should be regarded as of higher value than the model fit in evaluating to what extent the model reflects the hypothesized theory [24].

A subjective measure of sociodemographic status has formerly been validated through the use of the Mac Arthur Scale of Subjective Social Status, and found to reveal information not detected by using objective measures of sociodemographic affiliation [18,53].

### 4.5. Limitations

Even if SEM is considered a potentially powerful tool, it comes with additional statistical costs [54]. Unlike regression analyses, in SEM, all connected variables are assumed to display a linear association, which does not necessarily reflect reality. Moreover, SEMs are highly dependent on the correct specification of variables and the association between them. This possible source of error grows with the complexity of the SEM and is especially apparent for cross-sectional data, where the use of mediation is debated because causality cannot be addressed. Further, separate models for boys and girls were not built, and thus, the results cannot be used in addressing if there are gender-related differences in how health-related behaviors mediated the association between SSS and BMI.

Self-reported data including weights and heights were used to calculate BMI and to define overweight and obesity. Although other standard measures include waist circumference and skinfolds, BMI is recommended when conducting research at a population level [55]. The self-reported data provide a potential risk of random errors and therefore an underestimation of effect sizes and a lower explained variability by our models. This will affect the power to identify associations and consequently increase the likelihood of type 2 errors: In other words, reduce the likelihood of observing existing associations. Last, the initial ambition of including more latent variables might have been achievable if the questionnaire had originally been designed for the purpose of SEM analyses and included a more detailed recall of dietary intake. The model also lacks a variable displaying sedentary behavior. Thus, the complex field of adolescent behavior and obesity could have been be explored even more extensively.

### 4.6. Implications

The finding that health-related behaviors mediated the association between SSS and BMI in adolescents implies that sociodemographic status influences adolescent BMI through health-related habits. The results thus further suggest that sociodemographic status influences how adolescents are affected by the obesity-promoting society of today. Thus, prevention should aim at broad interventions targeting attitudes considering healthy behavior. It is also essential to develop strategies to prevent OWOB and ensure that these strategies do not enhance the social inequalities in health.

## 5. Conclusions

In this repeated cross-sectional study of 15–16-year-old Norwegian adolescents, perceived lower sociodemographic status was found as a risk factor for increased BMI in 2017 but not in 2002. This finding could be interpreted through the epidemiological transition. The association between SSS and BMI in adolescents was mediated through exercise as well as cigarette smoking, snuff-use, and alcohol-drinking. This finding can contribute to the understanding of the complexity of the association between sociodemographic status and OWOB in adolescents.

## Figures and Tables

**Figure 1 ijerph-17-07307-f001:**
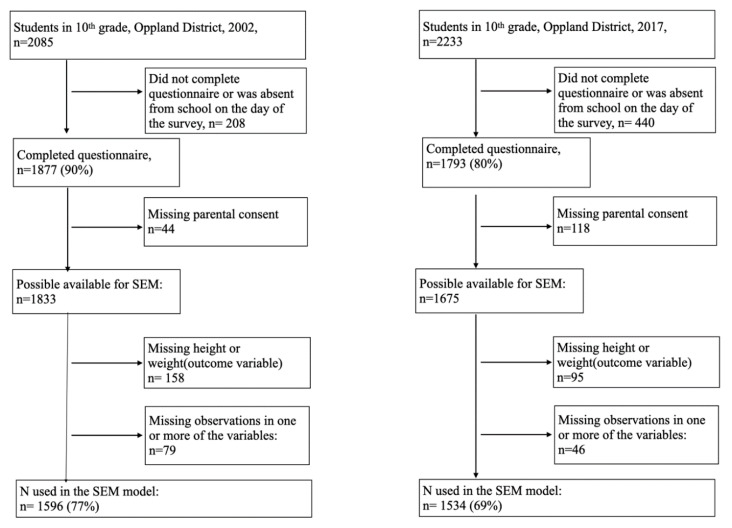
Flowchart of study-sample.

**Figure 2 ijerph-17-07307-f002:**
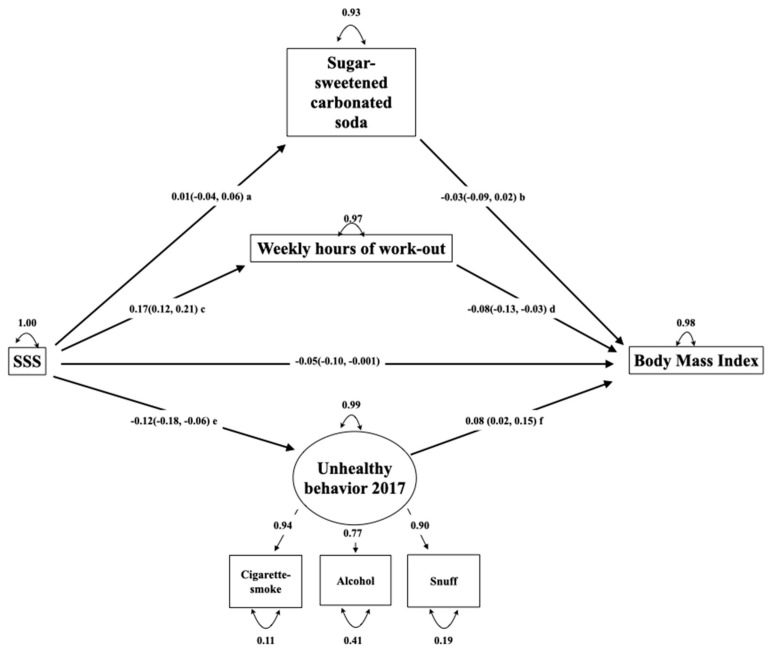
SEM for the 2017 survey including the latent variable. Numbers on straight arrows in the structural model indicate standardized β with 95% confidence intervals. Numbers on straight arrows in the measurement model indicate standardized regression coefficients between the latent variable and the observed variable. Numbers on top of curved arrows indicate unexplained variance. The rectangles indicate observed variables. The circle indicates a latent variable. Letters a, b, c, and d indicate parts of the indirect pathways with standardized β with 95% confidence intervals. The indirect pathways from SSS to body mass index: a*b = mediated by consumption of sugar-sweetened carbonated sodas = 0.001 (0.007, −0.001). c*d = mediated by weekly hours of workout -0.013 (−0.02, −0.004). Mediated effect: 20.6%. e*f = mediated by unhealthy behavior −0.009 (−0.002, −0.04). Mediated effect: 15.3%.

**Table 1 ijerph-17-07307-t001:** Characteristics of tenth-grade students (15–16yearold) in lower secondary schools in the district of Oppland, Norway.

	2002 n = 1596	2017 n = 1534
**Boys**	818 (51.3)	734 (47.8)
**Age**; years, mean (SD ^a^)	15.9 (0.3)	15.8 (0.3)
**Weight**; kg, mean (SD)	62.4 (11.1)	63.5 (11.4)
**Height**; cm, mean (SD)	171.6 (8.5)	171.9 (8.6)
**BMI**^b^; mean (SD)	21.1 (3.0)	21.4 (3.1)
**SSS** ^c^		
Poor	57 (3.6)	62 (4.0)
Average	620 (38.9)	469 (30.6)
Good	842 (52.8)	832 (54.2)
Very good	77 (4.8)	171 (10.7)
**Sodas** ^d^		
Seldom/never	169 (10.6.)	437 (28.5)
1–6 glasses weekly	837 (52.4)	893 (58.2)
1 glass daily	234 (14.7)	111 (7.2)
2–3 glasses daily	235(14.7)	70 (4.6)
≥4 glasses daily	121 (7.6)	23 (1.5)
**Smoking** ^e^		
Never	993 (62.2)	1336 (87.1)
Used to, but quit	151 (9.5)	92 (6.0)
Occasionally	236 (14.8)	97 (6.3)
Daily	216 (13.5)	9 (0.6)
**Snuff** ^f^		
Never	1337 (83.8)	1310 (85.4)
Used to, but quit	78 (4.9)	89 (5.8)
Occasionally	142 (8.9)	75 (4.9)
Daily	39 (2.4)	60 (3.9)
**Alcohol** ^g^		
No	209 (13.0)	650 (42.4)
Yes	1397 (87.0)	884 (57.6)
**Exercise** ^h^		
0	150 (9.4)	149 (9.7)
1–2	362 (22.7)	346 (22.6)
3–4	417 (26.1)	308 (20.1)
5–7	362 (22.7)	352 (22.9)
8–10	190 (11.9)	227 (14.8)
≥11	115 (7.2)	152 (9.9)

Data are presented as n (%) unless indicated otherwise. ^a^ SD = Standard deviation. ^b^ BMI = Body Mass Index, kg/m^2^. ^c^ SSS = Subjective Social Status, i.e., perceived family economy. ^d^ Consumption of sugar-sweetened carbonated sodas. ^e^ Cigarette smoking. ^f^ Use of snuff. ^g^ Ever drunk alcohol. ^h^ Hours of weekly leisure-time work-out.

**Table 2 ijerph-17-07307-t002:** Crude association between SSS ^a^ and BMI ^b^ in 15–16-year-old adolescents.

SSS ^a^	2002	2017
Poor	1.23 (0.42, 2.03)	0.03 (−0.80, 0.85)
Average	0 (Reference)	0 (Reference)
Good	0.04 (−0.27, 0.34)	−0.48 (−0.84, −0.13)
Very good	0.17 (−0.53, 0.87)	−0.58 (−1.13, −0.04)

Data collected in the District of Oppland, Norway. ^a^ SSS = subjective social status, i.e., perceived family economy. ^b^ BMI = Body Mass Index. Data are presented as the regression coefficient (95% confidence interval).

**Table 3 ijerph-17-07307-t003:** Spearman’s correlation coefficient (rho) for the included variables in the SEM for the 2017 survey, n = 1534.

	BMI ^a^	SSS ^b^	Soda ^c^	Smoking ^d^	Snuff ^e^	Alcohol ^f^
BMI ^a^						
SSS ^b^	−0.06 *					
Soda ^c^	−0.03	−0.02				
Smoking ^d^	0.05	−0.08 **	0.13 ***			
Snuff ^e^	0.04	−0.06 *	0.13 ***	0.68 ***		
Alcohol ^f^	0.05	−0.08 **	0.15 ***	0.31 ***	0.32 ***	
Exercise ^g^	−0.05	0.15 ***	−0.06 *	−0.09 ***	−0.07 **	−0.07 **

^a^ BMI = Body Mass Index, kg/m^2^. ^b^ SSS = Subjective Social Status, i.e., perceived family economy. Coded as poor–very good (4 categories). ^c^ Sugar-sweetened carbonated sodas. Coded as never–≥4 glasses daily (5 categories). ^d^ Cigarette smoking. Coded as no–daily (4 categories). ^e^ Snuff-use. Coded as no–daily (4 categories). ^f^ Ever tried alcohol. Coded as no/yes (2 categories). ^g^ Hours of weekly leisure-time work-out. Coded as 0–≥11 h weekly. * *p* ≤ 0.05; ** *p* < 0.01; *** *p* < 0.001.

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
