# Peer review of "Health-Related Behaviors in Adolescents Mediate the Association between Subjective Social Status and Body Mass Index"

_ijerph, 2020, doi:10.3390/ijerph17197307_

Round 1
Reviewer 1 Report
This well-written manuscript reports analyses of survey data collected from adolescents in 2002 and 2017 in one district of Norway with the goal of determining whether the association of subjective social status (SSS) and BMI was mediated by health-related behaviors. The investigators found no binary association of these variables in the 2002 data, so did not report further analyses. The investigators found a binary association in the 2017 data, and developed a model that showed the association was partially mediated by exercise and an unhealthy behaviors latent variable composed of alcohol use (never/ever), tobacco use, and snuff use.
I have some questions and concerns, with some concerns being relatively minor and others being more significant.
Questions
- Why, in the preliminary description of the results, is BMI the only variable that is mentioned as changing from 2002 to 2017 (on lines 165-168)? Inspection of Table 1 shows that the distribution of SSS appears to have shifted to be centered higher, and that for most of the behaviors, respondents reported healthier behaviors. If a preliminary comparison of 2017 to 2002 is going to be reported, I think this information should be included (unless there is some reason not to).
- Does any of what is reported vary with sex? Does the same model apply equally to girls and to boys? Were any analyses conducted that were contingent on sex?
- On lines 256 and following, the authors distinguish (correctly in the language of SEM) between direct and indirect associations (or effects) between SSS and BMI, but what does it mean for SSS to directly affect BMI? To the extent that the association is real, doesn’t all of the relationship between SSS and BMI have to be mediated by something (e.g., exercise, behaviors, diet, genes)? Maybe this question deserves some discussion.
- Lines 330-331: What would be the practical implementation of the suggestion that “sociodemographic differences be kept small to maintain normal weight in the majority of the population”?
Major Concerns
- It is difficult to determine, from this report, what the survey questions were, how many there were, and what, if anything, is known about the reliability and validity of these measures. I suggest that the survey be described more completely and that anything known about the reliability and validity of the questions be included. Maybe the survey questions could be included as supplementary material.
- In the same vein, I think that the description of the development of the measurement model could be improved. What variables were excluded? Was a model tested in which all behavioral variables were hypothesized to be indicators of one unhealthy behaviors latent variable? I assume that exercise and soda ended up as separate variables in the model because they didn’t load adequately on the unhealthy behaviors factor; is this true? How many models were evaluated? The sentence on line 138 can probably be deleted, given that there was only one latent variable—or it could be rewritten to say something like “We identified only one latent variable for which at least three observed variables were indicators”.
- Table 3 provides pairwise correlations for included variables. Perhaps as a supplementary table, the authors could show pairwise correlations for all variables, including those variables that ended up not being included in the SEM.
- In addition, somewhere the authors should explain some of the choices that were made for the response categories used. Why is there so much variation over questions in the grain of the response categories? For example, why is alcohol an never/ever question, whereas tobacco has multiple categories of ever?
Minor Concerns
- The abstract and introduction are written in way that presumes an association between SSS and BMI, but the investigators found SSS and BMI to be unrelated in the 2002 data. Leaving aside whether they knew this before collecting the 2017 data, the abstract and introduction should be written in a way that does not presume the association. (Given that there was no association in 2002, and given that there were 15 years between 2002 and 2017, one thinks that the absence of an association in the 2002 data would have been known by 2017.)
- Another suggestion for slight rewording: Although, obviously, different respondents completed the survey in 2002 and in 2017, someone not reading carefully might almost be confused into thinking that this was a longitudinal study in which the same people completed the survey twice. This could be solved easily by describing, in the abstract and the introduction, the procedure as something like “In each of 2002 and 2017, tenth graders in schools in Oppland were invited to complete a survey.”
- The caption of Figure 1 should be revised to indicate that it shows how the population was reduced to the analytic sample. I don’t have a good suggestion, but “Flowchart for the SEMs” doesn’t seem correct.
- On line 162, remove “except the SEM analyses”.
- On lines 218-220, the word “affects” might be removed. The paragraph could be rewritten as “In repeated cross-sectional surveys of 15-16 year-old Norwegians, we found an association between SSS and BMI in 2017, but not in 2002. In 2017, this association was mediated by frequency of exercise and unhealthy behavior (alcohol use, smoking, and snuff use).”
- Lines 230-231: Should a correlation of -.06, as found in this study, be described as supporting a “robust” association?
- Tables A.1 and A.2 are referred to in the text as Tables 4 and 5. There should be consistency between the text and the table labels.
- I did not check the references carefully, but in looking at Reference 43, noticed that some of the authors are missing. This should be corrected (and maybe the other references should be carefully checked).
Author Response
Dear Reviewer.
Thank you for your thorough review and comments. Our reply to your comments and questions follow the question or comment in italic letters.
Questions
- Why, in the preliminary description of the results, is BMI the only variable that is mentioned as changing from 2002 to 2017 (on lines 165-168)? Inspection of Table 1 shows that the distribution of SSS appears to have shifted to be centered higher, and that for most of the behaviors, respondents reported healthier behaviors. If a preliminary comparison of 2017 to 2002 is going to be reported, I think this information should be included (unless there is some reason not to).
- Thank you for this accurate observation. We have added the following sentences to the results-section: “In 2017, a higher percentage of adolescents reported having good and very good family economy, and a lower percentage of adolescents reported having average and poor economy.More adolescents also had healthier habits in 2017.”
- Does any of what is reported vary with sex? Does the same model apply equally to girls and to boys? Were any analyses conducted that were contingent on sex?
- On lines 256 and following, the authors distinguish (correctly in the language of SEM) between direct and indirect associations (or effects) between SSS and BMI, but what does it mean for SSS to directly affect BMI? To the extent that the association is real, doesn’t all of the relationship between SSS and BMI have to be mediated by something (e.g., exercise, behaviors, diet, genes)? Maybe this question deserves some discussion.
- Thank you for this notion. We have added the following text as the last paragraph under the title “the association between SSS and BMI in 2002 and 2017”: In general, several mediators can be included when investigating the association between SSS and BMI. Both differences in nutrition, physical activity, the built environment, and genes have been discussed [1-3]. It could also be speculated if the psychological resources in adolescents including social capital have a mediating role in the association between sociodemographic status and BMI, as found in adults [4,5].
- We have further included the following lines to the discussion: “The relationship between sociodemographic status and BMI is dynamic, and has been found as both positive and negative depending on the country`s developmental status [6]. Therefore, it seems reasonable to explore the societal frames for health-related choices for mediators of this association. Besides the finding of a direct association between SSS and BMI in 2017…….”
- Lines 330-331: What would be the practical implementation of the suggestion that “sociodemographic differences be kept small to maintain normal weight in the majority of the population”?
Thank you for this comment. We have decided to remove this last comment from the paper, as it disguised the sentence ahead that we consider to be more accurate.
Major concerns
- It is difficult to determine, from this report, what the survey questions were, how many there were, and what, if anything, is known about the reliability and validity of these measures. I suggest that the survey be described more completely and that anything known about the reliability and validity of the questions be included. Maybe the survey questions could be included as supplementary material.
Thank you for this comment. We used a Norwegian questionnaire, and we do not thibk it would be of value to you as it is. If you want this document translated to an English version, we can , of course, translate it. The questionnaire has not been formally validated, but a pilot-study was performed to investigate if the questions were written in an understandable language prior to the survey both in 2002 and 2017.
- In the same vein, I think that the description of the development of the measurement model could be improved. What variables were excluded? Was a model tested in which all behavioral variables were hypothesized to be indicators of one unhealthy behaviors latent variable? I assume that exercise and soda ended up as separate variables in the model because they didn’t load adequately on the unhealthy behaviors factor; is this true? How many models were evaluated? The sentence on line 138 can probably be deleted, given that there was only one latent variable—or it could be rewritten to say something like “We identified only one latent variable for which at least three observed variables were indicators”.
- Thank you for this valuable comment. The information regarding excluded variables can be found in Table 1 (all variables used in the models) and in Table A.1 (all variables explored).
- The latent variable was explored due to our hypothesis, and we explored three potential latent variables: a diet high in sugar (consumption of cakes and candy, sugar-sweetened carbonated soda, lemonade and energy drinks), level of exercise (hours of leisure-time workout, member of sports team and screen time), and unhealthy behavior (cigarette smoking, snuff use, alcohol drinking, and frequency of brushing teeth). We have added a sentence describing how we identified one latent variable: “The latent variable should also include at least three or more observed variables [24]. One such latent variable including the variables cigarette smoking, snuff-use, and alcohol drinking was identified.”
- We have added text to describe how many models we built in the paragraph describing the structural model: “After identifying one latent variable, exercise and consumption of sugar-sweetened soda was defined as mediators, as they represent central health related behaviors that are risk-factors for OWOB. Thus, only one structural model was built.”
- The suggested sentence is changed: “One such latent variable including the variables cigarette smoking, snuff-use, and alcohol drinking was identified.”
- Table 3 provides pairwise correlations for included variables. Perhaps as a supplementary table, the authors could show pairwise correlations for all variables, including those variables that ended up not being included in the SEM.
- Thank you for your suggestion, this table is now included as a supplementary table (Table A.2 in the Appendix B).
- In addition, somewhere the authors should explain some of the choices that were made for the response categories used. Why is there so much variation over questions in the grain of the response categories? For example, why is alcohol a never/ever question, whereas tobacco has multiple categories of ever?
Thank you. The questions and the categories were made by the National Public Health Institute as they made the survey prior to the 2002-survey. We aimed at including questions with more than 4 categories of answers, as this is most suitable when using SEM. As pointed out, “ever tried alcohol” has only two categories of answers. This question was chosen over another question, namely “have you ever been drunk” that had 5 categories of answers: never, once, 2–3 times, 4–10 times, >10 times. This choice was made as we according to our hypothesis considered it more accurate to address the health-related behavior of drinking alcohol or not, instead of the association regarding how many times the adolescent had been drunk. We have included a commentary in the second paragraph after Table 3: “Questions were preferably included if containing more than 4 categories of answers, as this is most suitable when using SEM [24]. Still, “ever tried alcohol” was chosen even if it had only two categories of answers. This choice was made as the question was considered accurate in addressing the health-related behavior of drinking alcohol or not.”
Minor Concerns
- The abstract and introduction are written in way that presumes an association between SSS and BMI, but the investigators found SSS and BMI to be unrelated in the 2002 data. Leaving aside whether they knew this before collecting the 2017 data, the abstract and introduction should be written in a way that does not presume the association. (Given that there was no association in 2002, and given that there were 15 years between 2002 and 2017, one thinks that the absence of an association in the 2002 data would have been known by 2017.)
Thank you for these considerations. The wording in the third paragraph has been changed accordingly: “The association between sociodemographic status and BMI in adolescence has been found as dynamic, and studies have revealed both non-existing, negative and positive associations [6,8,9]. During recent decades, food and drinks that are high in calories and low in nutrition have become more accessible [10]. Furthermore, the need for physically demanding work and transportation has declined [11]. These changes might have affected the association between sociodemographic status and BMI over recent decades [6]. However, the association between adolescent sociodemographic status and health outcomes including OWOB have been found complex, and in need of further elucidation [12].“
- Another suggestion for slight rewording: Although, obviously, different respondents completed the survey in 2002 and in 2017, someone not reading carefully might almost be confused into thinking that this was a longitudinal study in which the same people completed the survey twice. This could be solved easily by describing, in the abstract and the introduction, the procedure as something like “In each of 2002 and 2017, tenth graders in schools in Oppland were invited to complete a survey.”
- Thank you for this feedback. The wording in the abstract is changed into; “In 2002 (n=1596) and 2017(n=1534), tenth-grade students (15–16 years old) in schools in the Oppland County, Norway, completed a survey.”
- The wording in the last paragraph of the introduction section is changed into; “Overweight and obesity in adolescents is a global problem [19]. Thus, analyses should be done to understand more of the mechanisms that influence increased BMI in adolescents. The a priori hypothesis was that the association between SSS and BMI was mediated by health-related behaviors. Therefore, the aim was to investigate the possible relationship between SSS and BMI in adolescents at two different time points, and to determine whether the association was mediated by health-related behaviors in cross-sectional samples of 15–16-year-old adolescents. An analysis was performed to explore whether unhealthy nutrition, lower amount of exercise, and other unhealthy habits mediated the association between sociodemographic status and BMI.”
- The caption of Figure 1 should be revised to indicate that it shows how the population was reduced to the analytic sample. I don’t have a good suggestion, but “Flowchart for the SEMs” doesn’t seem correct.
Thank you for this suggestion. We have changed the name of Figure 1 into; “Flowchart of study-sample”.
- On line 162, remove “except the SEM analyses”.
Thank you, we have now removed these words from the manuscript.
- On lines 218-220, the word “affects” might be removed. The paragraph could be rewritten as “In repeated cross-sectional surveys of 15-16 year-old Norwegians, we found an association between SSS and BMI in 2017, but not in 2002. In 2017, this association was mediated by frequency of exercise and unhealthy behavior (alcohol use, smoking, and snuff use).”
Thank you for this comment. The wording is now changed, and due to other reviewers comments, also changed into third-person writing: “In repeated cross-sectional surveys of 15–16-year-old Norwegians, a linear association between SSS and BMI was found in 2017, but not in 2002. In 2017, this association was mediated by frequency of exercise and unhealthy behavior, including cigarette smoking, snuff-use and alcohol drinking.”
- Lines 230-231: Should a correlation of -.06, as found in this study, be described as supporting a “robust” association?
Thank you for this comment, we have changed the wording into “Accordingly, several nuances of the association between OWOB and SSS has been revealed, and the association seems valid.”.
- Tables A.1 and A.2 are referred to in the text as Tables 4 and 5. There should be consistency between the text and the table labels.
Thank you for this comment, all tables are now mentioned by the correct name.
- I did not check the references carefully, but in looking at Reference 43, noticed that some of the authors are missing. This should be corrected (and maybe the other references should be carefully checked).
Thank you for this comment, the references are now formatted using the MDPI style available online.
- Mekonnen, T.; Havdal, H.H.; Lien, N.; O'Halloran, S.A.; Arah, O.A.; Papadopoulou, E.; Gebremariam, M.K. Mediators of socioeconomic inequalities in dietary behaviours among youth: A systematic review. Obesity Reviews 2020.
- Evans, G.W.; Jones-Rounds, M.L.; Belojevic, G.; Vermeylen, F. Family income and childhood obesity in eight European cities: the mediating roles of neighborhood characteristics and physical activity. Social Science & Medicine 2012, 75, 477-481.
- Tyrrell, J.; Wood, A.R.; Ames, R.M.; Yaghootkar, H.; Beaumont, R.N.; Jones, S.E.; Tuke, M.A.; Ruth, K.S.; Freathy, R.M.; Davey Smith, G., et al. Gene–obesogenic environment interactions in the UK Biobank study. International journal of epidemiology 2017, 46, 559-575, doi:10.1093/ije/dyw337.
- Claassen, M.A.; Klein, O.; Bratanova, B.; Claes, N.; Corneille, O. A systematic review of psychosocial explanations for the relationship between socioeconomic status and body mass index. Appetite 2019, 132, 208-221, doi:10.1016/j.appet.2018.07.017.
- Mackenbach, J.D.; Lakerveld, J.; van Oostveen, Y.; Compernolle, S.; De Bourdeaudhuij, I.; Bárdos, H.; Rutter, H.; Glonti, K.; Oppert, J.M.; Charreire, H., et al. The mediating role of social capital in the association between neighbourhood income inequality and body mass index. Eur J Public Health 2017, 27, 218-223, doi:10.1093/eurpub/ckw157.
- Broyles, S.; Denstel, K.; Church, T.; Chaput, J.; Fogelholm, M.; Hu, G.; Kuriyan, R.; Kurpad, A.; Lambert, E.; Maher, C. The epidemiological transition and the global childhood obesity epidemic. International Journal of Obesity Supplements 2015, 5, S3-S8.
- Kline, R.B. Principles and practice of structural equation modeling; Guilford publications: 2015.
- West, P. Health inequalities in the early years: is there equalisation in youth? Social science & medicine 1997, 44, 833-858.
- Bann, D.; Johnson, W.; Li, L.; Kuh, D.; Hardy, R. Socioeconomic inequalities in childhood and adolescent body-mass index, weight, and height from 1953 to 2015: an analysis of four longitudinal, observational, British birth cohort studies. The Lancet. Public health 2018, 3, e194-e203, doi:10.1016/s2468-2667(18)30045-8.
- Swinburn, B.; Sacks, G.; Vandevijvere, S.; Kumanyika, S.; Lobstein, T.; Neal, B.; Barquera, S.; Friel, S.; Hawkes, C.; Kelly, B. INFORMAS (I nternational Network for Food and Obesity/non‐communicable diseases Research, Monitoring and Action Support): overview and key principles. Obesity reviews 2013, 14, 1-12.
- Katzmarzyk, P.T.; Mason, C. The physical activity transition. Journal of Physical Activity and Health 2009, 6, 269-280.
- Viner, R.M.; Ozer, E.M.; Denny, S.; Marmot, M.; Resnick, M.; Fatusi, A.; Currie, C. Adolescence and the social determinants of health. The Lancet 2012, 379, 1641-1652.
Reviewer 2 Report
Congratulations. This study contributes with new and important findings to the field of the investigation.
Comments
General comment
The authors used, in almost all the manuscript, a personal format. We measured (line 26), We found (line 29). I suggest, for example, to use a not personal format, It was measured (line 26), It was found (line 29).
Specific comments
Line 34 – To start with ...”In conclusion, in this…
Line 76 – I suggest to add a hypothesis at the end of the Introduction section.
Line 123-124- to put this hypothesis also at the end of the Introduction “hypothesis was that the association between SSS and BMI was mediated by health related behaviors”
Line 138 – to improve the presentation of this reference …variables (19) (page 201)…I suggest to transfer (page 201) to the reference section. To check in similar presentation of the page in other lines (294, 296, 300)
Table 1 – See also in the Tables of the Appendix A and B
-I suggest to use “body mass” instead of “weight”, here and in all the manuscript.
-The authors must include the unity of the BMI
-The authors must clarify about the volume of the glasses (200, 300 ml???)
I suggest to add references to aid in the Discussion and Introduction section. Please, see below:
1: Heikkala E, Ala-Mursula L, Taimela S, Paananen M, Vaaramo E, Auvinen J, Karppinen J. Accumulated unhealthy behaviors and psychosocial problems in adolescence are associated with labor market exclusion in early adulthood – a northern Finland birth cohort 1986 study. BMC Public Health. 2020 Jun 5;20(1):869. doi: 10.1186/s12889-020-08995-w. PMID: 32503491; PMCID: PMC7275307.
2: Underwood JM, Brener N, Thornton J, Harris WA, Bryan LN, Shanklin SL, Deputy N, Roberts AM, Queen B, Chyen D, Whittle L, Lim C, Yamakawa Y, Leon-Nguyen M, Kilmer G, Smith-Grant J, Demissie Z, Jones SE, Clayton H, Dittus P. Overview and Methods for the Youth Risk Behavior Surveillance System - United States, 2019. MMWR Suppl. 2020 Aug 21;69(1):1-10. doi: 10.15585/mmwr.su6901a1. PMID: 32817611.
3: Kyung Y, Lee JS, Lee JH, Jo SH, Kim SH. Health-related behaviors and mental health states of South Korean adolescents with atopic dermatitis. J Dermatol. 2020 Jul;47(7):699-706. doi: 10.1111/1346-8138.15386.
4: Xiang B, Wong HM, Perfecto AP, McGrath CPJ. The association of socio-economic status, dental anxiety, and behavioral and clinical variables with adolescents' oral health-related quality of life. Qual Life Res. 2020 Sep;29(9):2455-2464.doi: 10.1007/s11136-020-02504-7
5: Cheon YM, Ip PS, Haskin M, Yip T. Profiles of Adolescent Identity at the Intersection of Ethnic/Racial Identity, American Identity, and Subjective Social Status. Front Psychol. 2020 May 15;11:959. doi: 10.3389/fpsyg.2020.00959. PMID: 32499743; PMCID: PMC7244255.
6: Rahal D, Chiang JJ, Fales M, Fuligni AJ, Haselton MG, Slavich GM, Robles TF. Early life stress, subjective social status, and health during late adolescence. Psychol Health. 2020 May 13:1-19. doi: 10.1080/08870446.2020.1761977. Epub ahead of print. PMID: 32400197.
7: Steen PB, Poulsen PH, Andersen JH, Biering K. Subjective social status is an important determinant of perceived stress among adolescents: a cross-sectional study. BMC Public Health. 2020 Apr 16;20(1):396. doi: 10.1186/s12889-020-08509-8
Author Response
Dear Reviewer.
Thank you for your valuable comments.
- The personal format is now changed to a third person format.
- The suggested change in line 34 is made. The new wording is;
- “In conclusion, a direct association between SSS and BMI was found in 2017 in this repeated cross-sectional survey of 15–16 year-old Norwegian adolescents. This association was mediated through health-related behavior.”
- The suggested change in line 76 is made. The new wording of the last paragraph is
- “Overweight and obesity in adolescents is a global problem [19]. Thus, analyses should be done to understand more of the mechanisms that influence increased BMI in adolescents. The a priori hypothesis was that the association between SSS and BMI was mediated by health-related behaviors. Therefore, the aim was to investigate the possible relationship between SSS and BMI in adolescents at two different time points, and to determine whether the association was mediated by health-related behaviors in cross-sectional samples of 15–16-year-old adolescents. An analysis was performed to explore whether unhealthy nutrition, lower amount of exercise, and other unhealthy habits mediated the association between sociodemographic status and BMI.»
- The suggested change in line 123 and 124 is made according to the suggestion in the review.
- The pagination of the references is changed according to the suggestions made by the reviewer.
- Tables:
- Thank you for your consideration regarding use of body mass instead of weight. We consider the word body mass to be a more correct description, but the word weight is probably more readable and more easily understandable. As this is a paper covering public health issues, it is important to address also non-academic readers, and we have chosen not to exchange weight for body mass.
- The unity of BMI has been included in all tables.
- The volume of the glasses was regrettable not more clearly specified in the questionnaire. Thus, we have used it as a measure of frequency of consumption, not volume. We also think that most adolescents are not aware of how many milliliters their average glasses can hold, and thus “glasses” will be a familiar measure of how often they poor themselves a glass of e.g., soda.
- Thank you for suggesting these interesting papers to aid in our discussion. As far as we can see, however, these do not directly address overweight and obesity in adolescents nor any of the other related issues that we discuss in the discussion part of our paper. In case we overlooked something, we would highly appreciate more specific comments on how to include them.
Reviewer 3 Report
My question is the following, how can an association be established between variables measured in different years? I think the measurement in such a wide range 2002 and 2017 is a big limitation of the study. Obviously, the socioeconomic conditions of the subjects were not the same and it can influence the variables related to health. Even the study is carried out in a rural area and the limitations have not been taken into account, this can be a great limitation. Nor were the subjects the same. I advise focusing on a certain time, the most recent. Otherwise, other statistical analyzes would have to be made to determine the difference of means in both times. Therefore, I am considering rethinking the study focusing on a single year.
Some important considerations.
Initials or acronyms should not appear in the title.
The introduction lacks justification. No data are provided indicating the prevalence or incidence of Overweight and Obesity. It is not specified which studies exist to justify the research. The statistical tests with which the study is carried out should not appear in the introduction. I get the feeling that the important thing about the study is the SEM, and this is just a statistical analysis model without more.
Methodology: There are many shortcomings at the ethical level, procedure, collection instruments. In the statistical analysis, what level of significance was used? Non-parametric tests have been used, have normality tests been performed?
Table 2, the data provided are significant.
The discussion and conclusions of this study should be deepened.
Author Response
- My question is the following, how can an association be established between variables measured in different years? I think the measurement in such a wide range 2002 and 2017 is a big limitation of the study. Obviously, the socioeconomic conditions of the subjects were not the same and it can influence the variables related to health.
Thank you for these comments. We have analyzed the associations separately for the two surveys, and only built a model for the 2017-data. Thus, the possibly changed socioeconomic status of the subjects was not compared, but analyzed in their context. Further, the finding of a linear association between SSS and BMI in 2017 but not in 2002, reveals sociodemographic status as a risk factor for increased BMI in 2017, but not in the societal frames of 2002.
- Even the study is carried out in a rural area and the limitations have not been taken into account, this can be a great limitation.
Thank you for addressing this limitation. Oppland County is a mainly rural area when compared to other counties in Norway. Nevertheless, Norway is considered an egalitarian welfare-state, with small geographical differences. We have included a sentence regarding this issue in the paragraph 2.1 Protocol and subjects: “Although Oppland is a predominantly rural county, Norway is considered an egalitarian welfare state with relatively small differences between the counties.”
- Nor were the subjects the same.
Thank you for addressing this issue. We have studied the association between SSS and BMI in 10th grade students, aged 15–16 years in Oppland County at two different timepoints. This repeated cross-sectional study explores if SSS was a risk factor for increased BMI in two populations that grew up in the same geographical surroundings, but with possible different environments.
- I advise focusing on a certain time, the most recent.
Thank you for this comment. In the building of a model for 2017, the focus in the paper lays on the most recent time, and thus the associations in the present-day society.
- Otherwise, other statistical analyzes would have to be made to determine the difference of means in both times.
Thank you for this comment. We absolute agree with the reviewer, if we were to compare the two datasets, other statistical analyses should have been used. However, that is not what we have done.
6.Therefore, I am considering rethinking the study focusing on a single year.
Thank you for this comment, please see the answer above “I advise focusing on a certain time, the most recent. “
7. Some important considerations. Initials or acronyms should not appear in the title.
Thank you. We have changed the title into using Body Mass Index instead of BMI.
- The introduction lacks justification. No data are provided indicating the prevalence or incidence of Overweight and Obesity.
Thank you for this comment. We have included the prevalence of overweight an obesity in adolescents in Norway in the introduction section. Further, the last two paragraphs of the introduction have been changed.
- It is not specified which studies exist to justify the research.
Thank you. We have added more text and 5 new references to the introduction-section: [1-5].
- The statistical tests with which the study is carried out should not appear in the introduction.
Thank you for pointing this out. The wording in the last paragraph of the introduction has been changed into: “Overweight and obesity in adolescents is a global problem [5]. Thus, analyses should be done to understand more of the mechanisms that influence increased BMI in adolescents. The a priori hypothesis was that the association between SSS and BMI was mediated by health-related behaviors. Therefore, the aim was to investigate the possible relationship between SSS and BMI in adolescents at two different time points, and to determine whether the association was mediated by health-related behaviors in cross-sectional samples of 15–16-year-old adolescents. An analysis was performed to explore whether unhealthy nutrition, lower amount of exercise, and other unhealthy habits mediated the association between sociodemographic status and BMI.”
- I get the feeling that the important thing about the study is the SEM, and this is just a statistical analysis model without more.
Thank you for this notion on the methodology used in the paper. As overweight and obesity is a threat to present and future public health, we think that new information should be sought through not only linear, but also conceptual methodology. Low sociodemographic status is an established risk factor for this public health problem, and the use of SEM allows to explore several associations including the use of latent variables. This method has its limitations, as mentioned in the limitations-section of the paper, but it also enables us to explore the observed covariation in the indicator variables of the latent variable and possibly understand more of how these behaviors are related to SSS.
Methodology:
a)There are many shortcomings at the ethical level
Thank you for addressing the ethical concerns of this paper. The regional ethics committee acknowledged the study in both 2002 and 2017. We also collected consent from the parents of all participants younger than 16 years of age, and all participants older than 16 years of age are allowed to sign themselves according to the Norwegian law. Further, repeated measurements of height and weight are included in the routine healthcare for children and adolescents in Norway. As no established treatment or prevention program exists, we feel that research aiming to provide more knowledge of risk factors for overweight and obesity are also supported ethically.
b), There are many shortcomings procedure, collection instruments.
Thank you for this comment. Our data are self-reported by the adolescents, and the limitations are addressed in the limitations section: “Self-reported data including weights and heights were used to calculate BMI and to define overweight and obesity. Although other standard measures include waist circumference and skinfolds, BMI is recommended when conducting research at a population level [6]. The self-reported data provide a potential risk of random errors and therefore an underestimation of effect sizes and a lower explained variability by our models. This will affect the power to identify associations and consequently increase the likelihood of type 2 errors: In other words, reduce the likelihood of observing existing associations.”
c)In the statistical analysis, what level of significance was used?
Thank you. We set the level of significance to 5%, although we did not interpret this as a definite cut-off, in line with the current statistical and epidemiological understanding of the issue [7]. We have included this information into the text, in the paragraph 2.2 Statistical analyses.
d) Non-parametric tests have been used, have normality tests been performed?
Thank you for this comment. As the dataset included several ordinal variables, we used the WLSME procedure, that takes into consideration that the variables are ordinal. In addition, we explored if the outcome variable BMI was normally distributed, as also noted in the manuscript, “An approximately normal distribution was found for BMI (skewness and kurtosis 1.21 and 2.81 for 2002, 1.14 and 2.44 for 2017).”
13. Table 2, the data provided are significant.
Thank you for this comment. The crude analyses for 2002 regarding mean BMI by group of SSS reveals a statistically significant increased mean BMI for the lowest group of family economy, poor, compared to the reference category, average family economy. This poor-category included 57 individuals, whilst there were 620, 842 and 77 in the other categories. The categories good and very good perceived family economy did not reveal increased mean BMI using average family economy as an indicator in the numbers from 2002. Thus, we interpret the numbers as not revealing a linear association. This was also found when exploring the association between SSS and BMI in 2002: (standardized ß -0.02, [95% confidence interval (CI) -0.07,0.03]).
- The discussion and conclusions of this study should be deepened.
Thank you for this notion. We have included a paragraphn considering possible mediators of the association between SSS and BMI. We have also included two more paragraphs under the headline “unhealthy behaviors in adolescents” in the discussions-section. The limitations section is broadened, and the implications section is rewritten.
- Pulgaron, E.R. Childhood obesity: a review of increased risk for physical and psychological comorbidities. Clinical Therapeutics 2013, 35, A18-A32.
- Singh, A.S.; Mulder, C.; Twisk, J.W.; van Mechelen, W.; Chinapaw, M.J. Tracking of childhood overweight into adulthood: a systematic review of the literature. Obesity Reviews 2008, 9, 474-488, doi:10.1111/j.1467-789X.2008.00475.x.
- Adler, N.E.; Epel, E.S.; Castellazzo, G.; Ickovics, J.R. Relationship of subjective and objective social status with psychological and physiological functioning: Preliminary data in healthy, White women. Health Psychology 2000, 19, 586.
- Bradshaw, M.; Kent, B.V.; Henderson, W.M.; Setar, A.C. Subjective social status, life course SES, and BMI in young adulthood. Health Psychology 2017, 36, 682.
- Azzopardi, P.S.; Hearps, S.J.C.; Francis, K.L.; Kennedy, E.C.; Mokdad, A.H.; Kassebaum, N.J.; Lim, S.; Irvine, C.M.S.; Vos, T.; Brown, A.D., et al. Progress in adolescent health and wellbeing: tracking 12 headline indicators for 195 countries and territories, 1990-2016. Lancet (London, England) 2019, 393, 1101-1118, doi:10.1016/S0140-6736(18)32427-9.
- Hall, D.M.B.; Cole, T.J. What use is the BMI? Archives of disease in childhood 2006, 91, 283-286, doi:10.1136/adc.2005.077339.
- Greenland, S.; Senn, S.J.; Rothman, K.J.; Carlin, J.B.; Poole, C.; Goodman, S.N.; Altman, D.G. Statistical tests, P values, confidence intervals, and power: a guide to misinterpretations. European Journal of Epidemiology 2016, 31, 337-350, doi:10.1007/s10654-016-0149-3.
Reviewer 4 Report
- Suggest review first person language throughout - this reviewer's preference would be to avoid the subjective pronouns 'we' and 'our' that predominate and instead follow a more academic writing third person style - however this will depend on editorial office preference
- Suggest review parenthesis use - an number of times there double parentheses are evident (e.g. lines 30, 31 33, 191) - suggest instead to use brackets inside parentheses to create a double enclosure in the text. Avoid parentheses within parentheses, or nested parentheses.
- Suggest rephrase to avoid starting sentences with an abbreviation (e.g. line 171, 226, 307)
- Line 158 - suggest review citation of R here (see https://intro2r.com/citing-r.html for correct way to cite this software)
- References - there are some inconsistencies in style - some journal titles abbreviated, others not; inconsistencies with journal title capitalization evident; a couple of article titles are capitalized (18, 30) which is variant to majority
Author Response
Dear Reviewer.
Thank you for your comments, revisions have been made accordingly:
- The subjective pronouns "we" and "our" have been changed, using third-person style.
- The use of parenthesis has been changed, brackets are used inside parenthesis to avoid double parenthesis.
- Sentences have been rephrased to avoid starting with abbreviations.
- Citation of R has been changed.
- The references have been updated to have consistency in style as suggested.
Round 2
Reviewer 3 Report
The authors have made the indicated modifications.